# Quality of adolescent and youth-friendly health services in Dehana district public health facilities, northeast Ethiopia: Using the Donabedian quality framework

**Muluye Gebrie[1], Geta Asrade[2], Chalie Tadie Tsehay[2], Lake Yazachew[2], Endalkachew Dellie** [2]*

**1** Dehana District Health Office, Wag Hemra Zone, Amhara National Regional State, Dehana, Ethiopia, **2** Department of Health Systems and Policy, Institute of Public Health, College of Medicine and Health Sciences, University of Gondar, Gondar, Ethiopia

* endalkd.07@gmail.com

## Abstract

### Introduction

Adolescents and youths who need exceptional healthcare are the shapers and leaders of our global future. However, many of them are died prematurely, while others suffer from diseases partly because of the poor quality of health services. Thus, this study aimed to assess the quality of Adolescent and Youth-friendly Health Services (AYFHS) and associated factors in the public health facilities of Dehana district, Northeast Ethiopia.

### Methods

A facility-based quantitative cross-sectional study supported with a qualitative component was conducted from February 24 to March 30, 2020. The quality of AYFHS was measured using the Donbidean framework (structure, process and output component). Accordingly, a total of 431 adolescents and youths, five health facilities, twenty-five client-provider interaction observations, and nine key informant interviews were conducted. Binary logistic regression analysis was done, and variables with a p-value of < 0.05 and Adjusted Odds Ratio (AOR) with its 95% Confidence Interval (CI) were used to measure the association. The qualitative data were audio-recorded and transcribed verbatim. Then, Open Code 4.03 software was used to manage and analyze the data.

### Results

The quality of adolescent and youth-friendly health services was 58.8, 46.4, and 47.2% for structural, process, and output quality dimensions, respectively. The predictor variables for output quality were, being a student (AOR: 2.07, 95%CI: 1.07–3.40), farmers (AOR: 2.59, 95%CI: 1.25–5.39), own income sources (AOR: 1.99, 95%CI: 1.03–3.85), exempted services (AOR: 2.30, 95%CI: 1.43–3.71) and long waiting time (AOR: 3.8495%CI: 1.80–8.23).

**Data Availability Statement:** All data generated or analyzed during this study are included in this paper and its supplementary information files.

**Funding:** The University of Gondar has sponsored this study. However, it does not have a role to play in the design of the study and the collection, analysis and interpretation of the data and the writing of the manuscript.

**Competing interests:** All authors declared that they have no competing interest.

**Abbreviations:** AOR, adjusted odds ratio; AYFHS, adolescent and youth-friendly health service; CI, confidence interval; WHO, World Health Organization.

## Conclusions

The overall quality of adolescent and youth-friendly health services was still lower than the WHO good quality standards. The structural quality dimension was affected by the unavailability of adequate and trained health service providers, poor engagement of adolescents and youths in the facility governance structure, unavailability of guidelines, protocols and procedures. In contrast, the process quality dimension was also compromised due to the provider's poor compliance with the national AYFHS guidelines. Therefore, health facilities need to engage adolescents and youths in the health facility governance structure, and providers should comply with the national guideline.

## Background

Adolescents and youths are the shapers and leaders of our global future who need exceptional healthcare [1]. However, many adolescents and youths die prematurely, many suffer from diseases, and others may be ignored during healthcare, which deters their ability to grow and develop to their full potential [2, 3]. This is partly because adolescents and young people are more exposed to risky health behaviours, such as unsafe sex, alcohol, and drug abuse [3].

The World Health Organization (WHO) defined adolescents and youths as people between the ages of 10–19 and 15–24 years, respectively, and these two overlapping age groups as "young people" in the 10–24 year age groups [4]. There are around 1.8 billion adolescents and youths in the world today, 90% of whom live in developing countries. Of these, 16.08% were adolescents (10–19 years of age), and 15.5% are youths (15–24 years of age) [1, 5]. Thus, adolescents and youths make up 33.8% of the country's population [6].

Over 1.5 million adolescents and youths aged 10–24 years died globally in 2019, mainly from preventable or treatable causes [7]. In low and middle-income countries of Africa, adolescents' mortality rates were nearly thirteen times higher than in high-income countries and over seven times among youths [8]. The 2016 Ethiopia Demographic and Health Survey (EDHS) estimated that youth mortality rates were 11.61 deaths per 1000 population [9]. Injuries (including road traffic injuries and drowning), self-harm, violence, and maternal conditions are the top causes of death among adolescents and youths [10, 11].

Providing good quality health services for adolescents and youths has substantial implications. On the contrary, poor quality of health services is a significant driver of mortality across different health issues [12]. Low-quality health services associate with a range of sexual and reproductive health issues that can have undesirable consequences for maternal and child health [3, 13, 14]. Thus, it is essential to give good quality healthcare to promote adolescents and youths' health, wellness, and development [3].

However, evidence from both high and low-income countries showed that healthcare quality for adolescents and youths remains poor [3, 15–18]. For example, in Southern Ethiopia, the quality of AYFHS in public health facilities was 54.4, 42.0, and 49.1% for structural, process, and output quality dimensions, respectively [19]. Lack of privacy and confidentiality, poor attitudes towards providers, unavailability of the necessary equipment to offer the essential service package such as health information materials, essential drugs, and supplies, inadequate and untrained service providers are some of the contributing factors of poor quality of services [3, 20, 21].

Ethiopia has developed the first adolescent sexual reproductive health strategic plan (2007–2015) to discuss young sexual reproductive health needs in its development strategy [22].

Furthermore, in 2016, the Federal Ministry of Health (FMOH) established a comprehensive national strategy on adolescent and youth health (2016–2020). The goal of the strategy was to cut adolescent and youth mortality by 50% [6]. One of the priorities to meet this goal was enhancing fair access to high-quality, efficient and successful AYFHS [6].

Global health organizations such as WHO and the Joint United Nations Programme on HIV/AIDS have called countries to start standards-driven approaches to improve quality healthcare for adolescents [23]. WHO also recommends that governments offer high-quality, age-appropriate health services for the young population [24]. Prioritizing quality is a way of strengthening human rights-based approaches to healthcare. Recognizing this, more than 25 countries, including Ethiopia, have adopted national quality standards [3].

In 2009 the integrated family health program and the FMOH of Ethiopia jointly set up AYFHS in public health centers in line with its policies and strategies. Since then, the initiative has offered a range of services in separate corners of the health facilities [25].

However, limited studies were conducted to measure the quality of adolescent and youth-friendly health services using the Donabedian quality framework in Ethiopia. The Donabedian quality framework's advantage is its robustness to be applied to all healthcare settings and at many levels of the healthcare delivery system [26, 27]. Therefore, this study aimed to assess the quality of youth-friendly health services and associated factors in Dehana district public health facilities using the Donabedian quality framework.

Thus, the study fills the knowledge gap in the quality of AYFHS in the three dimensions and provides evidence for context-specific decision-making. Besides, it helps health care providers, policy-makers, and different organizations to improve AYFHS in public health facilities.

## Materials and methods

### Study design and settings

A facility-based quantitative cross-sectional study supported by a qualitative approach was conducted to assess adolescent and youth-friendly health service quality in Dehana district public health facilities, Waghemra zone, Amhara National Regional State, Ethiopia. The study was conducted from February 24 to March 30, 2020.

The district is located in northeast Ethiopia, 735 and 245 Kilometers away from Addis Ababa and Bahir Dar (the capital city of Amhara National Regional State). The Dehana district has a total of 147,073 inhabitants. Of which, 75,743 were women, and 4,207 were urban inhabitants. Of the total population, 49,686 were constituted by adolescents and youths. The district has one primary hospital, six health centers, and thirty-one health posts. Among these health facilities, five health centers provide AYFHS in separate classes.

All young people aged 10–24 years in the district were the target population. All young people aged 10–24 years who visited AYFHS during the study period in public health facilities in the Dehana district were the study population. For the qualitative part, health center managers and AYFHS providers were the study population.

AYFHS clients with emergency medical conditions and adolescents under 15 years of age who came alone to health facilities were excluded from the study.

### Sample size and sampling technique

The sample size for the survey was determined by using a single population proportion formula (n = (Zα/2)2*P (1-P)/(d)2). A confidence level (CI) of 95% and a margin of error (d) of 5% was used. From a similar study in Arba Minch town, Ethiopia, 49.1% of clients satisfied

with services provided was used [19], and adding 10% for non-response rate. Thus, the final sample size was obtained to be 418.

Moreover, five health facilities were observed for the structural quality assessment, and 25 observations were made to assess the client-provider interaction (three to five interactions at each site between health care providers and adolescent clients, as recommended by WHO) [28].

All the five health facilities which deliver AYFHS were included in the study. The sample was allocated proportionally to each public health facility based on the average number of young clients flow in the last one month. The study participants were then selected in every other client exit by using systematic random sampling techniques.

For key informant interviews, five health center heads were selected purposively to get rich information on the availability of resources and health facility management issues related to AYFHS. Besides, four AYFHS providers were included to obtain information on resources, client-provider interactions, and other AYFHS quality issues. In addition, Customer-provider interaction observation was performed by the principal investigator and experienced health providers from outside the study area based on the number of AYFHS providers in each health facility until the recommended number of observations was reached.

## Variables and measurements

The quality of AYFHS was assessed using the Donabedian quality framework of input, process, and output quality components [27]. In addition, the input (the structural component) quality of AYFHS was measured using 27 structural measurement items adapted from the WHO and FMOH standard tool [28, 29]. They were used to assess the adequacy of service providers, essential drugs, medical equipment and supplies, information, and basic infrastructures to provide the service.

The process AYFHS quality was measured using 17 items adapted from the WHO tool and other related literature was used [19, 30]. It was used to assess whether the interaction between client and providers was confidential, good communication, education and utilization of job aids, guidelines, and examination and treatment procedures are performed following the WHO standard.

Moreover, the output quality was measured by clients' satisfaction towards the services provided at AYFHS using 16 items questions. Further, each item question was outlined as a five-point Likert scale (1: strongly disagree to 5: strongly agree) [31].

Then for each component, when the health center scored 75% and above of the WHO quality criteria, classified as "good quality" or "good standard of care" 40–75%, classified as " medium quality of care" and one or more of the quality components scored below 40%, classified as "poor quality of care" [19, 30].

Waiting time was measured by self-report of young clients from the arrival at the health facility until getting the services. At the same time, the latrine was stated as clean when it had no faeces on or near the seats of latrines and when flies did not disturb clients. Similarly, clean water availability was declared as available when there was a piped or well-protected water in the health center.

## Data collection tools and procedures

The survey data were collected using an interviewer-administered structured questionnaire, while the key informant interviews used semi-structured questionnaires. The key informant interview guide, facility audit, and client-provider interaction observation checklists were adapted from the WHO [28] and national guidelines [29, 32]. The tools were initially

developed in English and translated into the local (Amharic) language, and then back to the English language by the language expert to make sure of its consistency and accuracy.

Five diploma nurses for data collection and three public health officers for supervision were employed. The principal investigator has observed the structure of health facilities, client-provider interaction and performs key informant interviews. One day of training was provided for both data collectors and supervisors on interviewing techniques, handling ethical issues, maintaining confidentiality and privacy. The overall internal reliability of the quantitative survey tools was checked using Cronbach's alpha reliability test, and the score for the items of each component was greater than 0.81. To ensure the study's internal validity, the tool was pre-tested on 21 clients at the Asketema health center in the Gazgibla district (the nearby districts). The principal investigator prepared the guiding questions by considering existing literature on AYFHS quality and experts (program experts and researchers) who have more than 3 years of work experience in the area were reviewed. Besides, a pre-test was done for the interview guiding questions by the same data collectors who collect the final data.

Moreover, the first and last three client-provider interaction observations were dropped per health care provider to lower the Hawthorn effect. In addition, the audio recorded data from key informant interviews were compared with transcribed written notes before being translated into English for completeness and accuracy, and the transcribed notes were read to key informants to assess accuracy and completeness throughout the transcriptions process. Finally, the qualitative data were analyzed based upon the information gained from the participants.

## Data management and analysis

The quantitative data were cleaned and checked for consistency, coded, entered into Epi-Data version 3.1 software, and then exported to SPSS version 20 software for analysis. Descriptive measures were computed to summarize the participants' socio-demographic characteristics and the quality of AYFHS. Bivariable and multivariable logistic regression analysis was used to assess any association between each independent variable and client satisfaction. Independent variables with a p-value of less than 0.2 during the bi-variable logistic regression were entered into the multiple logistic regression analysis. Finally, AOR with 95% CI and a P-value of less than 0.05 was used to identify statistically significant variables to the outcome variable.

The audio-recorded qualitative interviews were transcribed verbatim and translated from Amharic into English by the first author. The translations were also done by data collectors to ensure that the participants' original meanings were fully captured.The data were coded based on the preset categories developed by the Donabedian quality model for thematic analysis. Data management and analysis were done using Open Code 4.03 software. The first and last authors independently analyzed the data; this was confirmed by the third and fourth authors and validated by the second author. To support the quantitative findings, the findings were presented in three main themes in the form of narratives based on Donabedian quality of care dimensions (structural quality, process quality, and client satisfaction with AYFHS services).

## Ethical consideration

The ethical approval letter was obtained from the University of Gondar Research and Ethical Review Committee, College of Medicine and Health Sciences (Ref. No/IPH/837/06/2012). In addition, letters of authorization from the district health office and public health facilities were received before contacting participants. The participants were then fully briefed about the study's purpose and benefits and obtained informed written consent. Additionally, for respondents under 15 years old, oral assent from them and consent from their parents/guardians were obtained before collecting the data. Confidentiality was maintained through anonymity,

and privacy measures were taken to preserve the right of the respondent. Finally, the selected participants were asked about their willingness to join the study.

## Results

### Socio-demographic characteristics of participants

A total of 413 adolescent and youth-friendly health service users have participated in this study, with a response rate of 98.8%. About 57.4% of the participants were females. The mean age of respondents was 19.29 (SD ±3.36) years. The majority of participants (68.5%) were single, 79.9% were rural residents, and 77.5% did not have their income. Furthermore, nearly half of the participants (49.9%) were students, and 34.1% were secondary and preparatory school students (Table 1).

### Use of service and experiences

The finding showed that about 61.3% of the AYFHS users had frequently visited health facilities in the last twelve months. Of these, 58.1% visited 2–4 times. Most of the information to visit the health facilities was obtained from health workers (62.2%). Most clients received treatment services for illness-related conditions 279 (67.6%) followed by 76(18.4%) family planning services. Most clients received all the services they needed on the day of their visit (97.8%). 2.2 percent of service users, however, did not receive the services they wanted. The main reason for missing the services was the lack of medicines/supplies (55.6%). None of the two conditions that need referrals to higher health facilities was referred out with referral sheets.

Regarding information status, 77.7% of clients heard information about what services they could get in the facilities. But only 54.7% of adolescents and youths said basic services are

**Table 1. Socio-demographic characteristics of study participants in adolescent and youth-friendly health service in the Dehana district, northeastern Ethiopia, May 2020 (n = 413).**

| Characteristic | Category | Frequency (n) | Percent (%) |
|---|---|---|---|
| Sex | Male | 176 | 42.6 |
| | Female | 237 | 57.4 |
| Age in years | 10–14 | 36 | 8.7 |
| | 15–19 | 181 | 43.8 |
| | 20–24 | 196 | 47.5 |
| Marital status | Unmarried | 295 | 71.4 |
| | Married | 118 | 28.6 |
| Educational status | No education | 111 | 26.9 |
| | Primary school | 121 | 29.3 |
| | Secondary and preparatory | 141 | 34.1 |
| | College and above | 40 | 9.7 |
| Occupational status | Student | 206 | 49.9 |
| | Merchant | 18 | 4.4 |
| | Gov't employee | 32 | 7.7 |
| | Unemployed | 58 | 14.0 |
| | Farmer | 99 | 24.0 |
| Residence | Rural | 330 | 79.9 |
| | Urban | 83 | 20.1 |
| Having their own income | No | 320 | 77.5 |
| | Yes | 93 | 22.5 |

provided to youths. In addition, 85.5% of clients know where to go or whom to ask when they need services not provided in facilities.

## Accessibility and acceptability of the service

To most clients, the opening hours were convenient (89.8%), and the facilities were clean for (89.6%) service users. Similarly, most clients were comfortable with the sex of AYFHS providers. The finding of key informant interviews supports this result;

> *"Most clients are comfortable with the sex of health care providers, but often female clients with sexually transmitted infections prefer a female health care provider. For instance, three days ago, a girl with a sexually transmitted infection refused me when I asked her to do a physical examination"* (35 years old male AYFHS provider).

About 97.6% of clients recommended that other young people visit the facilities for AYFHS. Service quality (36.3%), friendly service providers (42.2%), affordable services (20.8%), free services (16.2%), short waiting times (13.3%), and comfortable compound and same-sex service providers (4.8%) were the preferred reasons for recommending the use of services. Clients who did not recommend other young people to visit the health facilities for AYFHS stated that unfriendly service providers and long waiting-times were the major contributing factors.

Concerning clients' payment status, 71.2% of service users paid for services, and 28.8% were received services freely. A key informant interview supports this finding as:

> *"Health facilities have to offer youth-friendly health services free of charge because health facilities do not cover the service charge because of budget constraints. Especially for those who use family planning services, they pay for laboratory urine tests that are costly for clients that do not have their source of income.*" (35 years old AYFHS provider)

> "*Since the reimbursement mechanism for exempted services is inconvenient, health facilities charge clients for pregnancy tests. Thus, contraceptive users face difficulties of paying for a pregnancy test to rollout pregnancy.*" (27 years old AYFHS provider)

About the time taken to reach the health facilities, 46% travel less than 30 minutes, 25.2% travel between 30 minutes and 1 hour, and 28.8% travel more than 60 minutes to reach the health facilities.

Related to the length of waiting time, 66.6% of clients waited less than 30 minutes, 23.2% waited 30 minutes to one hour, and 10.2% of clients waited more than one hour to obtain services after reaching health facilities.

> "*The wait time is not in the range of the national protocol due to a shortage of health personnel. One adolescent and youth-friendly health service provider give all the services provided in AYFHS rooms. But the AYFHS has two rooms, one room for family planning service and one room for outpatient service. Thus, when the health provider offers services in the family planning room, clients in the outpatient room wait until the health provider finishes tasks in the family planning room. Similarly, when the health provider offers services in the outpatient diagnosis room, clients in the family planning services room wait until the health provider finishes tasks in the outpatient room.*" (a health center head aged in the mid-20s)

## Structural quality

The overall adolescent and youth-friendly health service structural quality dimension scores were 58.8%. The maximum scores were from the Chila health center (64.1%), and the minimum scores were from the Arbit health center (54%).

Four health facilities had signboards referring to services available at AYFHS classes and opening hours for services about the signboard. In four health facilities, the service opening hours were from Monday to Friday (from 8:30 am- 11:30 pm). However, one health facility provides services seven days a week and 24 hours a day.

All facilities had a separate space for providing services to young people, and four health facilities had a separate waiting area for adolescents and youths. In addition, three health facilities have separated the consultation corner by screens to protect clients' privacy.

*"Young people do not want anyone to see them when they get a service, but we do not have a curtain to make rooms comfortable for clients.*" (head of the health facility, aged 29).

Three health centers had specially developed information materials for adolescents and young people in the waiting area.

In relation to the necessary competency for the required service packages, AYFHS is provided by trained health care providers in all health facilities. But there is insufficient training for AYFHS providers in which only one health worker is trained in each health facility. For example, only one health facility (Chila health center) had a service provider trained in compressive abortion care. In addition, in all health facilities, only one provider is assigned in the AYFHS unit. All service providers are males aged between 25 to 35 years.

Key informant interview results indicated that AYFHS providers were inadequate for the provision of minimum packages of services.

"*There is no sufficient number of trained AYFHS providers. Whenever this trained AYFHS provider is unavailable for any reason, we face a difficulty of getting health providers to offer the service replacing the trained AYFHS providers."* (A 27 years old male AYFHS provider)

*"In our health facility, healthcare staff turnover is acceptably high. Particularly trained and experienced health providers left our health facility and now we only have one trained AYFHS provider."* (A 28-year-old health center head)

Regarding the availability of necessary equipment to give the required packages, all health facilities had functional blood pressure measurement, stethoscope, fetal stethoscope, clinical thermometer, adult weighing scale, pregnancy test strip, height meter, and refrigerator. Only one health facility had an ophthalmoscope set, but three health centers had an otoscope set and computer with internet access. Communication devices are only used in one health center. None of the health facilities had a light source (torch) for examination. Three health facilities had urine test strips with ten parameters, and two health centers had Hemoglobinometers.

All health facilities had the necessary medical supplies and contraceptives. But in the three health facilities, there were no emergency contraceptive pills. Necessary drugs were available in all health facilities except magnesium sulfate., Paracetamol was available in the three health facilities.

Key informant interview results indicated no shortage of necessary drugs and supplies needed to provide AYFHS.

*"There is no shortage of drugs and supplies. Different NGOs like save the children and transform primary healthcare units support the delivery of AYFHS in our facility.*" (A 28 years old health center head)

Related to basic amenities, all health facilities had functional and clean toilets with bins for waste disposal, general waste disposal, safe sharp storage, and clinical waste disposal. Nevertheless, none of the health facilities had functional hand hygiene facilities, and only one health facility had clean drinking water. In addition, three health facilities had electricity during working hours and transportation (ambulance).

Regarding the availability of policies, standard operating procedures (SOPs), and protocols, none of the health facilities had policies, SOPs, and protocols to ensure confidentiality, informed consent, a safe and friendly environment, privacy, and free or affordable services.

Besides, none of the health facilities had supportive supervision tools. AYFHS, sexually transmitted infection, and HIV counseling and testing guidelines were only available in three health facilities, and four had family planning guidelines. Surprisingly, no health facility had a guideline on comprehensive abortion care.

All health facilities provide the recommended minimum services packages, except the comprehensive abortion care services delivered only in one health facility. In addition, the key informant interview results indicated that protocols for essential services were not available.

*"Our facility does not have all the required guidelines and protocols. The district health office and the NGOs have not distributed to our health facility. Hence, we did not have a guideline for AYFHS."* (A 29 years old male health center head)

Regarding youth involvement, none of the facilities included youths in their governance structure (planning, monitoring, and evaluation of health service delivery). In addition, none of the health facilities had clear AYFHS operation budgets (Table 2).

## Process quality

In this study, the overall process quality score of AYFHS was 46.4%. A higher process quality score was seen in Chilla health center (48.2%) and lower in Azila health center (43.5%).

The finding of client-provider interaction observation showed that all clients were secured their auditory privacy, 80% had their visual privacy guaranteed, and just 12% had assured their confidentiality. Similarly, None of the AYFHS providers introduced themselves to clients to build a good rapport. On the other hand, 84% of clients listened with their providers' attention, and providers measured the vital sign of 36% of the client. Likewise, the providers gave 76% of clients sufficient time to counsel or consult as required to the issue. Only 20% of clients asked about their psychosocial history to identify different risk factors and problems.

Health care providers used job aids and case management guidelines for only 16% of service users. About 64% of clients were asked for approval before the examination/procedure was started, 44% of clients received clear information on the medical condition, and 28% got clear information on the management or treatment options. Similarly, 36% of clients asked preferences for the management or treatment options, 60% of clients received information on risk reduction and prevention methods, 32% were informed about service availability, and all service users received clear information on follow-up actions. Surprisingly, none of the health facilities was used audiovisual material to educate service users. (Table 3)

**Table 2. List of structural quality measuring items fulfilled by Dehana district public health facilities, Northeast Ethiopia, 2020.**

| Structural availability | Amdework HC (%) | Chila HC (%) | Silda HC (%) | Arbit HC (%) | Azila HC (%) | District average (%) |
|---|---|---|---|---|---|---|
| A sign board that mentions operating hours | 100 | 0 | 100 | 100 | 100 | 80 |
| separate space to provide services to youth | 100 | 100 | 100 | 100 | 100 | 100 |
| Separate waiting area for adolescents and youths | 100 | 100 | 0 | 100 | 100 | 80 |
| Information materials for young in waiting area | 100 | 100 | 0 | 0 | 100 | 60 |
| AYFHS class open 24 hrs./ day and 7days/week | 0 | 0 | 0 | 100 | 0 | 20 |
| Services delivered by trained providers | 100 | 100 | 100 | 100 | 100 | 100 |
| Trained outreach workers | 0 | 100 | 0 | 0 | 100 | 40 |
| Updated lists of services included in the package | 100 | 100 | 100 | 100 | 100 | 100 |
| Does the facility have a functional referral and feedback (back referral) system | 33 | 67 | 33 | 0 | 67 | 40 |
| Providers competencies provide the required package of services | 60 | 80 | 60 | 60 | 60 | 64 |
| Providers' obligations and clients right displayed in the facility | 25 | 75 | 0 | 0 | 0 | 20 |
| Up-to-date decision support tools | 83 | 83 | 33 | 17 | 67 | 57 |
| Tools for supportive supervision in AYFHS care | 0 | 0 | 0 | 0 | 0 | 0 |
| Budget for AYFHS program implementation | 0 | 0 | 0 | 0 | 0 | 0 |
| Board of the facility include adolescent and youth | 0 | 0 | 0 | 0 | 0 | 0 |
| Engage young people in service delivery | 0 | 0 | 0 | 0 | 0 | 0 |
| Availability of necessary equipment | 100 | 81 | 56 | 56 | 75 | 73 |
| Availability of necessary supplies | 100 | 100 | 100 | 100 | 100 | 100 |
| Availability of necessary contraceptives | 80 | 80 | 100 | 100 | 80 | 88 |
| Stock-outs of any of the contraceptives or supplies in the last one month | 67 | 83 | 83 | 83 | 67 | 77 |
| Availability of necessary drugs or equivalents | 100 | 100 | 100 | 100 | 80 | 96 |
| Stock-outs of any of the any of drugs or equivalent in the last one month | 87 | 80 | 93 | 80 | 80 | 84 |
| Availability of necessary basic amenities | 90 | 70 | 70 | 70 | 70 | 74 |
| visual and auditory privacy protecting features | 0 | 50 | 100 | 50 | 100 | 60 |
| Are the algorithms for STI treatment and HIV tests available or displayed in AYFHS class | 50 | 50 | 50 | 0 | 100 | 50 |
| Availability AYFHS service relevant documents (policies/guidelines/SOPs) | 0 | 0 | 0 | 0 | 0 | 0 |
| Offered recommended minimum AYFHS Packages? | 93 | 100 | 93 | 93 | 87 | 93 |
| **Overall structural quality percentage** | **60.6** | **64.1** | **51.4** | **54** | **63.5** | **58.8** |

**Note,** HC: Health Center.

**Output quality.** The output quality concerns the degree of clients' satisfaction with service provided at service units to adolescents and youths. Accordingly, the overall client satisfaction towards AYFHS in Dehana district public health facilities was 47.2%, with a 95% CI(42.1–52.1). However, the level of satisfaction was varying across facilities, which were high in Chila health center (63.2%) and low in Azila health center (30%) (Fig 1).

Around 40.3% of males and 52.3% of females were satisfied with the services provided at AYFHS units. The satisfaction level among 10–24, 15–19, and 20–24 age groups was 30.6%, 49.7%, and 48%, respectively. Regarding marital status, 42% were single, 55.9% were married, and 83.3% were divorced and satisfied with AYFHS units' services.

Related to educational status, the satisfaction level of unable to read and write, able to read and write, attending primary school, attending secondary schools and college and above was

**Table 3. List of process quality measuring items performed by Dehana district public health facilities, Northeast Ethiopia, 2020.**

| Services characteristics | Amdework HC (%) | Chila HC (%) | Silda HC (%) | Arbit HC (%) | Azila HC (%) | District average (%) |
|---|---|---|---|---|---|---|
| Visual privacy | 100 | 0 | 100 | 100 | 100 | 80 |
| Auditory privacy | 100 | 100 | 100 | 100 | 100 | 100 |
| Ensuring confidentiality | 0 | 20 | 20 | 20 | 0 | 12 |
| Introduce himself/herself | 0 | 0 | 0 | 0 | 0 | 0 |
| listen with attention | 80 | 80 | 60 | 100 | 100 | 84 |
| Measure vital signs (BP, T0, RR, PR) | 20 | 40 | 40 | 40 | 40 | 36 |
| Take any psychosocial history | 20 | 20 | 20 | 40 | 0 | 20 |
| Use job aids and case management guides | 20 | 40 | 20 | 0 | 0 | 16 |
| Provide sufficient time for counseling/ consultation as required for the problem | 80 | 80 | 60 | 80 | 80 | 76 |
| Ask permission before performing the examination/procedure | 80 | 100 | 60 | 40 | 40 | 64 |
| Provide clear information on the medical condition | 40 | 40 | 40 | 40 | 60 | 44 |
| Provide clear information on the management or treatment options | 20 | 40 | 20 | 20 | 40 | 28 |
| Ask preferences for the management/ treatment options | 60 | 40 | 20 | 20 | 40 | 36 |
| Provide information on risk reduction and prevention Methods | 60 | 80 | 60 | 60 | 40 | 60 |
| Use audio-visual materials | 0 | 0 | 0 | 0 | 0 | 0 |
| Provide clear information on follow-up actions | 100 | 100 | 100 | 100 | 100 | 100 |
| Inform about services available for him/her | 20 | 40 | 60 | 40 | 0 | 32 |
| **Overall percentage** | **47.1** | **48.2** | **45.9** | **47.1** | **43.5** | **46.4** |

Note; HC: Health Center.

51.5%, 70.0%, 47.9%, 42.6%, and 45.0% satisfied with services they got, respectively. The majority (55.9 percent) of clients who had their income were satisfied with AYFHS services.

About the characteristics of care, the majority of respondents were satisfied with treatment in a respectful manner (92%), the convenience of location (86.7%), length of time of consultation (83.8%), friendliness of health workers (82.1%) and convenience opening hours of service (80.6%) respectively. While 65.4% were satisfied with supportive staff's friendliness, 59.6% were satisfied with the information on risk reduction, and 40% were satisfied with the prevention and adequacy of psychosocial assessment (Table 4).

## Overall quality

In this study, the overall quality of adolescent and youth-friendly health services in the Dehana district for structural, process, and output quality dimensions were 58.8%, 46.44%, and 47.2%, respectively (Fig 2).

## Factors associated with adolescent and youth client satisfaction

In this study, five variables were identified as statistically significant with adolescent and youth-friendly health services' satisfaction.

Accordingly, students were 2.07 times more likely to be satisfied with AYFHS than unemployed clients (AOR: 2.07, 95%CI: 1.07–3.40). Similarly, farmer participants were 2.59 times more likely to be satisfied than unemployed clients (AOR: 2.59, 95%CI: 1.25–5.39). Besides, clients who had their income source were 1.99 times more likely to be satisfied than those who do not have their income source (AOR:1.99, 95%CI: 1.03–3.85).

Likewise, clients who were free of charge were 2.3 times more likely to be satisfied than those who paid for AYFHS services (AOR:2.30, 95%CI:1.43–3.71). Moreover, clients who

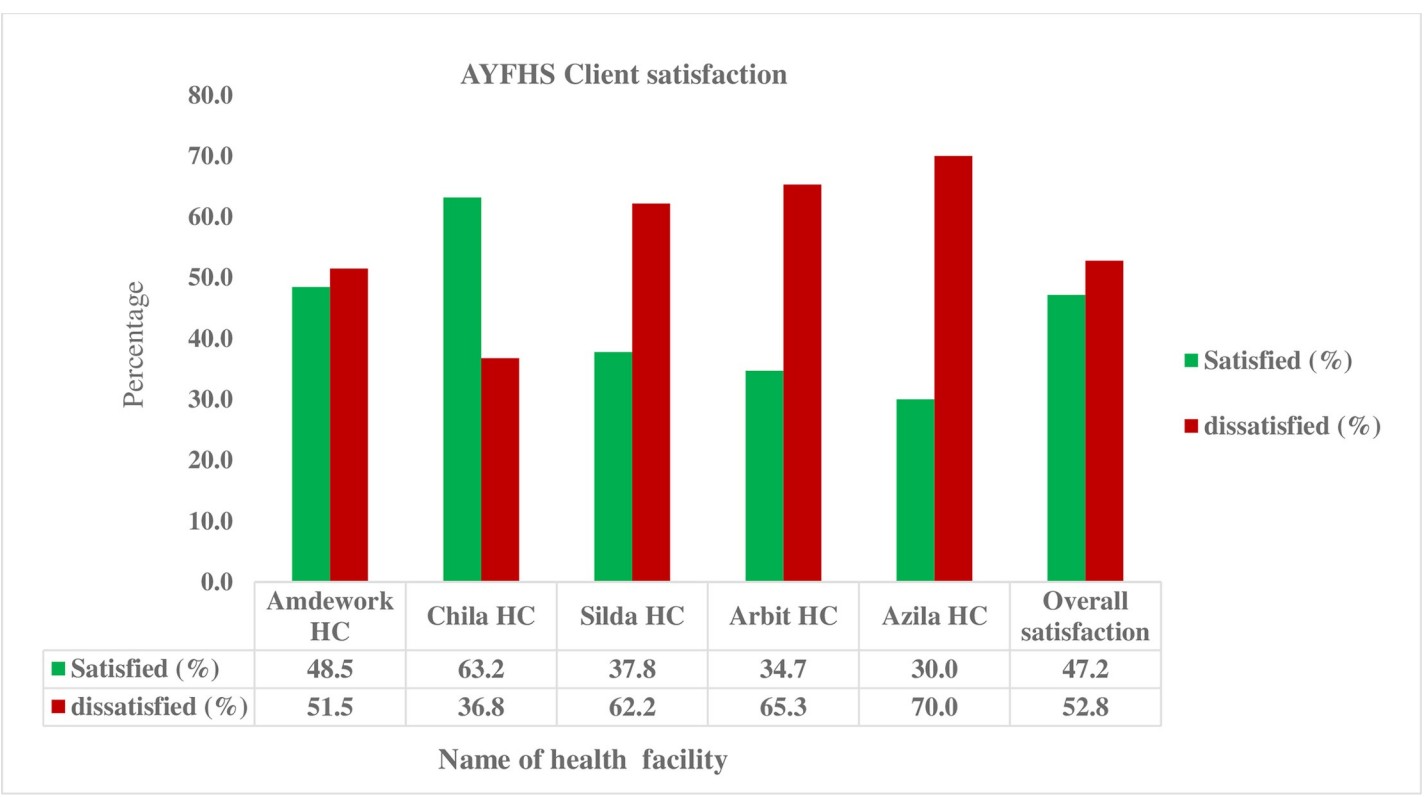

**Fig 1. Level of Satisfaction among adolescents and youths in Dehana district public health Northeast Ethiopia, 2020 (n = 413).**

**Table 4. Characteristics of care and satisfaction among respondents who received health services from Dehana district public health facilities, northeast Ethiopia, 2020 (n = 413).**

| Characteristics of care | Satisfied n(%) | Dissatisfied n(%) |
|---|---|---|
| The convenience of service opening hour | 333(80.9) | 80(19.4) |
| Length of waiting time | 279(67.6) | 134(32.4) |
| Friendliness of supportive staffs | 270(65.4) | 143(34.6) |
| Friendliness of health workers | 339(82.1) | 74(17.9) |
| Waiting area comfortableness | 307(74.3) | 106(25.7) |
| Privacy protection during the consultation | 276(66.8) | 137(33.2) |
| Length of time of consultation | 346(83.8) | 67(16.2) |
| Freedom of asking health care providers | 320(77.2) | 93(22.5) |
| Cost of services | 334(80.9) | 79(19.1) |
| Understanding of information given by health care provider | 323(78.2) | 90(21.8) |
| Treatment procedure | 306(74.1) | 107(25.9) |
| Adequacy of psychosocial assessment | 165(39.7) | 248(60.0) |
| Information is given on risk reduction and prevention | 246(59.3) | 167(40.4) |
| The convenience of the location of the AYFHS service delivery point | 358(86.7) | 55(13.3) |
| Treat in a respectful manner | 380(92.0) | 33(8.0) |
| Cleanliness of areas surrounding health facility | 333(80.4) | 80(19.4) |
| **Overall satisfaction** | **195(47.2)** | **218(52.8)** |

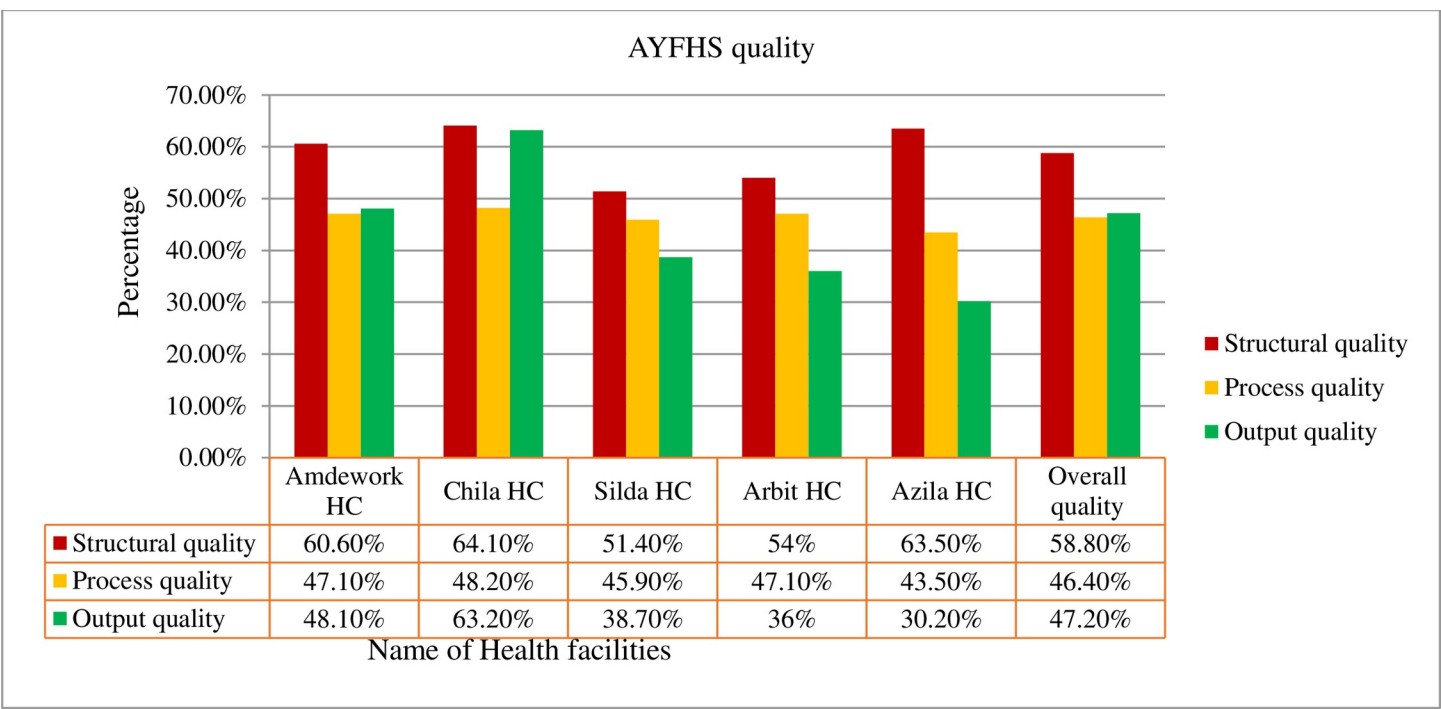

**Fig 2. Quality of adolescent and youth-friendly health services in Dehana district public health facilities, Northeast Ethiopia, 2020 (n = 413).**

waited less than 30 minutes were 3.84 times more likely to be satisfied than clients who waited over one hour (AOR: 3.8495%CI: 1.80–8.23) (Table 5).

## Discussion

In this study, the quality of AYFHS in Dehana district public health facilities was assessed using the Donabedian quality assessment framework. The findings showed that the structure, process, and output quality of AYFHS was 58.8, 46.4, and 47.2%, respectively.

This result is consistent with a similar study conducted in southern Ethiopia, which reported that services' structure, process, and output quality was 54.41, 42 and 49.10%, respectively [19]. However, in the west Gojjam zone, the three quality dimensions scored below 50% [33]. Similarly, it is consistent with that of a study conducted in the primary healthcare facilities of the northwest district of South Africa, in which the quality of AYFHS was 48% [20].

However, the finding is far below the WHO input, process, and output quality dimension standards, where a good quality score of 75% and above per quality dimension is required [3]. The difference could be due to the assessment of all structural quality standards of AYFHS in our study. However, the West Gojjam zone study assessed only the structural quality of reproductive health services for adolescent and youth clients.

Our study showed higher scores on drugs, supply, equipment, and basic amenities than other measures in all district's health facilities. This might be due to the fact that the general requirement to functions the healthcare facility as a whole, but not the entire show a good implementation of AYFHS. Besides, in all health facilities, services were delivered by trained health care providers in a separate space, and 80% of the health facilities have a particular waiting area. This might give comfort and maintain the privacy of service users. Likewise, a study conducted in Mexico indicated that a physical space allowing privacy during counselling is an essential element of quality AYFHS [34].

**Table 5. Bi-variable and multivariable logistic regression analysis of client satisfaction towards AYFHS in Dehana district public health facilities, Northeast Ethiopia, 2020 (n = 413).**

| Background characteristics | Satisfaction, n (%) | | COR (95%CI) | AOR (95%) |
|---|---|---|---|---|
| | **Satisfied** | **Dissatisfied** | | |
| Sex | | | | |
| Male | 71 (40.3%) | 105 (59.7%) | 0.62(0.42–0.91) | 0.86(0.54–1.38) |
| Female | 124 (52.3%) | 113 (47.7%) | 1 | 1 |
| Age in year | | | | |
| 10–14 | 11(30.6%) | 25 (69.4%) | 0.48(0.22–1.02) | 0.91(0.38–2.20) |
| 15–19 | 90(49.7%) | 91 (50.3%) | 1.07(0.72–1.61) | 1.32(0.80–2.18) |
| 20–24 | 94(48.0%) | 102 (52.0%) | 1 | 1 |
| Marital status | | | | |
| Unmarried | 129(43.7%) | 166(56.3%) | 0.61(0.40–0.94) | 0.80(0.47–1.36) |
| Married | 66(55.9%) | 52(44.1%) | 1 | 1 |
| Occupation | | | | |
| Student | 94(45.6%) | 112(54.4%) | 1.60(0.87–2.93) | 2.07(1.07–3.40) * |
| Farmer | 56(56.6%) | 43(43.4%) | 2.47(1.26–4.84) | 2.59(1.25–5.39) * |
| Merchant | 11(61.1%) | 7(38.9%) | 2.99(1.00–8.89) | 1.47(0.40–5.31) |
| Gov't employee | 14(43.8%) | 18(56.2%) | 1.48(0.61–3.58) | 0.99(0.35–2.85) |
| Unemployed | 20(34.5%) | 38(65.5%) | 1 | 1 |
| Own income | | | | |
| No | 143(44.7%) | 177(55.3%) | 1 | 1 |
| Yes | 52(55.9%) | 41(44.1%) | 1.57(0.99–2.50) | 1.99(1.03–3.85) * |
| Provider sex | | | | |
| Not comfortable | 51 (56.7%) | 39 (43.3%) | 0.62(0.38–0.99) | 0.81(0.48–1.38) |
| Comfortable | 144 (44.6%) | 179 (55.4%) | 1 | 1 |
| Payment status | | | | |
| Free | 75 (63.0%) | 44 (37.0%) | 2.47(1.60–3.83) | 2.30(1.43–3.71) * |
| Paying | 120 (40.8%) | 174 (59.2%) | 1 | 1 |
| Waiting time in minute | | | | |
| < = 30 | 156 (56.7%) | 119 (43.3%) | 3.69(1.78–7.65) | 3.84(1.80–8.23) * |
| 31–60 | 28 (29.2%) | 68 (70.8%) | 1.16(0.51–2.63) | 1.36(0.58–3.18) |
| >60 | 11 (26.2%) | 31 (73.8%) | 1 | 1 |
| Available service | | | | |
| Not informed | 38(41.3%) | 157(48.9%) | 1 | 1 |
| Informed | 54(58.7%) | 164(51.1%) | 1.36(0.85–2.18) | 1.16(0.65–2.07) |

* Statistically significant at a p-value of less than 0.05.

However, providing quality AYFHS is still a significant problem, and the health facilities were inefficient in fully adhering to AYFHS standards. In each health facility of the district, only one trained AYFHS provider was assigned. This could be due to a shortage of budget to train sufficient health care providers. This implies that the service provisions for service receivers were below the requirement. The result is comparable with those of studies carried out in the public health facilities of Arba Minch and West Gojjam zone, Ethiopia, which found that inadequate training for health workers was a major problem [19, 33].

This study also showed that the AYFHS was not available during the weekend and late afternoon. As most service users were students, it may be hard for young people to get health services if working hours coincide with school times. The findings are consistent with similar studies in Ghana and Arba Minch town, Ethiopia [19, 35].

None of the health facilities involved youths and adolescents in their governance structure (board) and during the planning, monitoring, and evaluation of healthcare service delivery, including AYFHS, which shows the absence of youths' engagements. This would make health services unresponsive to their needs. This is consistent with a study conducted in southern Ethiopia and reported that youths did not engage in healthcare governance structures [19].

Our study showed that all health facilities in the district have no policies, protocols, and Standard Operating Procedures (SOPs) on the AYFHS provision because they were not distributed from the district's health office. Lack of guidelines, protocols, and SOPs, combined with insufficient training in AYFHS, may significantly affect the quality of services. The finding is consistent with that of previous study findings, which showed that most facilities lack AYFHS policy, procedures, and protocols [36, 37].

The quality of AYFHS process is lower than the WHO good process quality standard [28]. This low process quality score may be attributed due to the insufficient training of AYFHS providers in the study area.

Similarly, the client-provider interaction result showed that 100% and 80% of clients' auditory and visual privacy were protected, respectively. The results are comparable with that of a study conducted in Arba Minch town, Ethiopia, showing that 100% and 66.6% of the clients are protected by auditory and visual privacy [19]. However, it is higher than a study conducted in the West Gojjam Zone, Ethiopia, which found that only 27.8% of health facilities provided visual and auditory privacy during the consultation [33]. This variation may be due to the availability of particular services provision rooms and waiting areas dedicated for AYFHS in Dehana district public health facilities that are essential for protecting service users' privacy.

Moreover, this study's process quality was mainly compromised by the inability of AYFHS providers to create good relationships with clients, assure confidentiality, and use audiovisual materials. The finding is consistent with those similar studies conducted in Ethiopia [19, 33]. Besides, the information provided on the availability of facility resources (32%), the risk reduction and prevention measures were relatively low (66%), and the psychosocial history assessment was inadequate (20%). Moreover, The finding is consistent with that of a study conducted in the West Gojjam Zone, Ethiopia, which reported inadequate psychosocial evaluations and information on the availability of services for young people [33].

Furthermore, overall client satisfaction with AYFHS was 47.2%. The finding is comparable with a similar study in Arba Minch town, Ethiopia, 49.1% [19]. However, the level of satisfaction is lower than that of studies conducted in South Africa (81.7%) [15], Dejen district, West Gojjam, Ethiopia, 60.7% [38] and Dessie town,58.9% [31]. The possible explanation for this disparity may be service delivery quality, subjective measures of satisfaction, and AYFHS client expectations.

Student and farmer participants were 2.07 and 2.59 times more likely to be satisfied with AYFHS provision than unemployed clients. The findings are consistent with the results of other similar studies in Ethiopia [19, 31]. Studies in Serbia showed that unemployed clients tend to perceive their health condition as worse and that the quality of the care they receive is low, which poses a barrier to contact health workers [18].

This study showed higher odds of client satisfaction among clients who received AYFHS for free compared to participants who paid for AYFHS services. This finding is consistent with a study done in the West Amhara region, Ethiopia [39]. This may be the fact that the financial capacity to pursue AYFHS and high medical costs may not be appropriate for adolescents and youths.

Also, clients who had their income source were 1.99 times more likely to be satisfied than their counterparts. This might be due to young people who have no their sources of income may not want to add a burden on their families by asking for extra money to pay for services. So they may hesitate to disclose that they need to obtain health services.

Moreover, clients who waited for less than 30 minutes to get the service were 3.84 times more likely to be satisfied than clients who waited for more than an hour. This finding is supported by a similar study done at different areas of the country; Jimma university specialized hospital [40], Southern Ethiopia [19], and the West Amhara region [39].

This could be attributed to the high client load and insufficient deployment of AYFHS providers in Dehana district public health facilities. It may also be due to poor awareness of service users recognizing that certain healthcare services require time to provide quality services.

## Strength and limitation of the study

The study was triangulated with various data sources and data collection methods to increase the findings' credibility and validity. Besides, the first and the last three client-provider interaction observations were dropped to minimize hawthorn's effect. Although participants were interviewed in separate areas, still social desirability bias may also be a factor.

## Conclusion

This study found that the overall quality of adolescent and youth-friendly health services was still lower than the expected WHO good quality standards of at least 75%. Structural quality was affected by the unavailability of adequately trained health workers and poor engagement of adolescents and youths in AYFHS provision at the health facilities. Besides, the unavailability of policies, protocols, procedures and important guidelines hinder structural quality.

Process quality was also compromised due to lack of assurance of client confidentiality; none of the providers introduced themself to build a good relationship with clients, non-use of audiovisual aids, and inadequate psychosocial history assessments. Besides, the given information on risk reduction and prevention methods, medical conditions, management options, and preferences of management options also affect the process quality of AYFHS. Being unemployed, paying for services, not having their income, and waiting time to get services were the predictor variables to client satisfaction (output quality).

Therefore, the district health office needs to prepare and distribute guidelines, protocols, procedures essential to AYFHS delivery. Also, provision of training to health care providers for the minimum service delivery packages of AYFHS, especially for safe abortion services, is not given in 80% of health facilities in the study area. Health facilities also need to engage adolescents and youths in the facility governance structure to plan, implement, and monitor activities. To reduce long waiting times for clients, adequate and trained providers should be assigned to healthcare facilities in AYFHS rooms, and the health care providers should comply with the national guidelines.

## Supporting information

**S1 File.**
(RAR)

## Acknowledgments

The authors would like to thank the Dehana district health office, health professionals working in district health centers, supervisors, and data collectors for their unreserved contribution to this study.

## Author Contributions

**Conceptualization:** Muluye Gebrie, Geta Asrade, Lake Yazachew, Endalkachew Dellie.

**Data curation:** Muluye Gebrie, Geta Asrade.

**Formal analysis:** Muluye Gebrie, Geta Asrade, Chalie Tadie Tsehay, Endalkachew Dellie.

**Funding acquisition:** Muluye Gebrie.

**Investigation:** Muluye Gebrie.

**Methodology:** Muluye Gebrie, Chalie Tadie Tsehay, Endalkachew Dellie.

**Project administration:** Muluye Gebrie.

**Resources:** Muluye Gebrie.

**Software:** Muluye Gebrie, Geta Asrade, Chalie Tadie Tsehay, Lake Yazachew, Endalkachew Dellie.

**Supervision:** Muluye Gebrie.

**Validation:** Muluye Gebrie, Geta Asrade, Chalie Tadie Tsehay.

**Visualization:** Muluye Gebrie, Lake Yazachew.

**Writing – original draft:** Chalie Tadie Tsehay, Lake Yazachew, Endalkachew Dellie.

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
