## [Decision Letter · Decision Letter 0]

9 Jun 2021

PONE-D-20-35107

Quality of adolescent and youth-friendly health services in Dehana district public health facilities, northeast Ethiopia: Using the Donabedian quality framework

PLOS ONE

Dear Dr. Dellie,

Thank you for submitting your manuscript to PLOS ONE. After careful consideration, we feel that it has merit but does not fully meet PLOS ONE’s publication criteria as it currently stands. Therefore, we invite you to submit a revised version of the manuscript that addresses the points raised during the review process.

We look forward to receiving your revised manuscript.

Kind regards,

Kingston Rajiah

Academic Editor

PLOS ONE

Journal Requirements:

2. Please include additional information regarding the quantitative survey or qualitative questionnaire used in the study and ensure that you have provided sufficient details that others could replicate the analyses. For instance, if you developed a questionnaire as part of this study and it is not under a copyright more restrictive than CC-BY, please include a copy, in both the original language and English, as Supporting Information, or include a citation if it has been published previously."

"In the Methods, please clearly discuss how the questionnaire was validated and pre-tested."

“In your Methods section, please provide additional information about the participant recruitment method and the demographic details of your participants. Please ensure you have provided sufficient details to replicate the analyses such as: a)  a description of any inclusion/exclusion criteria that were applied to participant recruitment, b) a statement as to whether your sample can be considered representative of a larger population, c) a description of how participants were recruited, and d) descriptions of where participants were recruited and where the research took place.

Reviewers' comments:

Reviewer's Responses to Questions

**Comments to the Author**

1. Is the manuscript technically sound, and do the data support the conclusions?

Reviewer #1: Yes

Reviewer #2: Partly

2. Has the statistical analysis been performed appropriately and rigorously? 

Reviewer #1: Yes

Reviewer #2: Yes

3. Have the authors made all data underlying the findings in their manuscript fully available?

Reviewer #1: Yes

Reviewer #2: Yes

4. Is the manuscript presented in an intelligible fashion and written in standard English?

Reviewer #1: Yes

Reviewer #2: Yes

5. Review Comments to the Author

Reviewer #1: Quality of adolescent and youth-friendly health services in Dehana district public health

facilities, northeast Ethiopia: Using the Donabedian quality framework

Manuscript Number: PONE-D-20-35107

Accept with no comments

Reviewer #2: Firstly, thank you for inviting me to review this article, it made for interesting reading. While the study is significant and the findings are interesting, I am unsure about the novelty of the manuscript and whether it adds more to the existing literature on the same topic of interest. That being said, it is a well written article and I applaud the authors for their efforts to increase awareness regarding a significant section of the population, the health care services available for adolescents and their inadequacies.

If similar studies using the same quality assessment tools have been conducted before, what is the justification for conducting this study? Is there a novel component to this study that merits publication and dissipation of findings to the scientific community?

The English could be improved in the introduction part of the article, there are several places where authors have missed out grammatical errors.

In the first paragraph of the introduction, the authors can mention common causes of mortality in adolescents and youth in addition to risky behaviors.

It would be more appropriate to mention 'target population' than 'source population' in the methods part of the article. This is just a suggestion for the authors to consider.

Was the investigator who collected qualitative data trained in qualitative research beforehand? What was the exact method used? Please mention in-depth interviews if that was the method chosen to collect qualitative data.

Was the validity and reliability of the Amharic translated version of the semi-structured questionnaire checked? If it was, please report the values in terms of Cronbach's alpha and other appropriate measures. If it was not, what is the justification for not checking the validity and reliability?

There are some grammatical errors in the discussion as well, authors can proof read and make corrections accordingly.

6. PLOS authors have the option to publish the peer review history of their article (what does this mean?). If published, this will include your full peer review and any attached files.

Reviewer #1: No

Reviewer #2: **Yes: **Nesa Aurlene

---

## [Author Response · Author response to Decision Letter 0]

26 Aug 2021

Dear Editor, 

Thank you for your comments and suggestions to improve our manuscript. All responses for the reviewers' and editor's comments and suggestions are included in the "Response to reviewers" file. 

Kindly review our clean version of the revised manuscript. 

Thank you!!!.

---

## [Decision Letter · Decision Letter 1]

5 Oct 2021

Quality of adolescent and youth-friendly health services in Dehana district public health facilities, northeast Ethiopia: using the Donabedian quality framework

PONE-D-20-35107R1

Dear Dr. Dellie,

We’re pleased to inform you that your manuscript has been judged scientifically suitable for publication and will be formally accepted for publication once it meets all outstanding technical requirements.

Kind regards,

Kingston Rajiah

Academic Editor

PLOS ONE

Additional Editor Comments (optional):

Reviewers' comments:

Reviewer's Responses to Questions

**Comments to the Author**

1. If the authors have adequately addressed your comments raised in a previous round of review and you feel that this manuscript is now acceptable for publication, you may indicate that here to bypass the “Comments to the Author” section, enter your conflict of interest statement in the “Confidential to Editor” section, and submit your "Accept" recommendation.

Reviewer #2: All comments have been addressed

2. Is the manuscript technically sound, and do the data support the conclusions?

Reviewer #2: Yes

3. Has the statistical analysis been performed appropriately and rigorously? 

Reviewer #2: Yes

4. Have the authors made all data underlying the findings in their manuscript fully available?

Reviewer #2: (No Response)

5. Is the manuscript presented in an intelligible fashion and written in standard English?

Reviewer #2: Yes

6. Review Comments to the Author

Reviewer #2: Dear authors, thank you for addressing all my comments and my concerns regarding the manuscript, I have gone through the revised manuscript and I find it acceptable for publication now. I thank you once again for inviting me to review your manuscript, I applaud your efforts to bring visibility to a public health issue of such relevance, and I wish you success in publishing this research.

7. PLOS authors have the option to publish the peer review history of their article (what does this mean?). If published, this will include your full peer review and any attached files.

Reviewer #2: **Yes: **Nesa Aurlene

---

## [Editor Report · Acceptance letter]

12 Oct 2021

PONE-D-20-35107R1 

Quality of adolescent and youth-friendly health services in Dehana district public health facilities, northeast Ethiopia: using the Donabedian quality framework 

Dear Dr. Dellie:

I'm pleased to inform you that your manuscript has been deemed suitable for publication in PLOS ONE. Congratulations! Your manuscript is now with our production department. 

Kind regards, 

on behalf of

Dr. Kingston Rajiah 

Academic Editor

PLOS ONE